# Fluorescent Quinolinium Derivative as Novel Mitochondria Probe and Function Modulator by Targeting Mitochondrial RNA

**DOI:** 10.3390/molecules28062690

**Published:** 2023-03-16

**Authors:** Bo-Zheng Wang, Ying-Chen Zhou, Yu-Wei Lin, Xiu-Cai Chen, Ze-Yi Yu, Yao-Hao Xu, Jia-Heng Tan, Zhi-Shu Huang, Shuo-Bin Chen

**Affiliations:** Guangdong Provincial Key Laboratory of New Drug Design and Evaluation, School of Pharmaceutical Sciences, Sun Yat-sen University, Guangzhou 510006, China

**Keywords:** RNA fluorescent probe, OXPHOS inhibitor, mitochondrial RNA, anticancer

## Abstract

Mitochondria have a crucial role in regulating energy metabolism and their dysfunction has been linked to tumorigenesis. Cancer diagnosis and intervention have a great interest in the development of new agents that target biomolecules within mitochondria. However, monitoring and modulating mitochondria RNA (mtRNA), an essential component in mitochondria, in cells is challenging due to limited functional research and the absence of targeting agents. In this study, we designed and synthesized a fluorescent quinolinium derivative, **QUCO-1**, which actively lit up with mtRNA in both normal and cancer cells in vitro. Additionally, we evaluated the function of **QUCO-1** as an mtRNA ligand and found that it effectively induced severe mitochondrial dysfunction and OXPHOS inhibition in RKO colorectal cancer cells. Treatment with **QUCO-1** resulted in apoptosis, cell cycle blockage at the G2/M phase, and the effective inhibition of cell proliferation. Our findings suggest that **QUCO-1** has great potential as a promising probe and therapeutic agent for mtRNA, with the potential for treating colorectal cancer.

## 1. Introduction

The Warburg effect is a metabolic adaptation observed in cancer cells where they rely more on glycolysis, even in the presence of oxygen, instead of oxidative phosphorylation (OXPHOS) as the primary source of energy [1,2]. This allows cancer cells to meet the high energy demands required for their rapid proliferation and growth. However, recent studies have shown that mitochondrial function, especially OXPHOS, is critical for cancer cell survival and growth [3,4]. OXPHOS not only is the powerhouse of cancer cells, but also plays a vital role in regulating various cellular processes such as apoptosis, autophagy, ferroptosis, and the cell cycle, which are closely linked to tumorigenesis.

Mitochondria play a central role in maintaining cellular homeostasis by regulating the production of reactive oxygen species (ROS), which can have both pro- and antitumorigenic effects. Due to the fatal effects of acute mitochondrial dysfunction on cancer cells, there is growing interest in discovering novel agents that target biomolecules within mitochondria and inhibit the OXPHOS functions of cancer cells [5]. A number of small molecules targeting the electron transport chain (ETC) and mitochondrial respiratory chain complexes have demonstrated success in cancerous models such as the biguanides metformin and phenformin as mitochondrial complex I inhibitors, and **VLX600** and tigecycline as different ETC inhibitors [6,7,8,9,10]. To validate the effect in a high-throughput way, the mitochondrial membrane potential (MtMp), a mitochondrial health indicator, is a good parameter. MtMp dysfunction probes have several types such as off/on probes and ratiometric fluorescent probes. Numbers of target-switchable fluorescent probes have been reported [11,12,13,14,15,16,17].

Mitochondrial RNA (mtRNA) has recently gained attention from researchers [18]. Mitochondria form the bioenergetic and biosynthetic organelle, containing its own DNA genome (mtDNA) that transcripts mtRNA, including mitochondrial tRNA, rRNA, unknown functional RNA, and mitochondrial mRNA [19]. mtDNA mainly codes for the mitochondrial respiratory chain complexes responsible for OXPHOS [20]. Although previous studies have focused on the connection between tumorigenesis and mtDNA mutation [21,22,23], the regulation of cancer cells by mtRNA remains unclear.

The study of mtRNA has been limited due to the lack of available methods. Currently, the only ways to visualize mtRNA in fixed cells are through BrU-based immunofluorescence and FISH using oligonucleotide probes [24,25]. To gain an insight into the biological function of mtRNA, there is a need to develop small molecule probes that can monitor or modulate mtRNA in live cells.

Previously, we successfully developed selective RNA fluorescent probes, **QUID-1** and **QUID-2**, to visualize RNA in live cells [26,27]. In the following evaluation of their analogs, we discovered a quinolinium-based coumarin hemocyanin probe, **QUCO-1**, that lit up with mtRNA, both in vitro and in live cells. Considering the connection between mitochondrial dysfunction and tumorigenesis, we further evaluated **QUCO-1** as an mtRNA ligand. We found it could modulate mitochondrial dysfunction and cause proliferation inhibition and cell death in RKO cells. These results suggest that **QUCO-1** could serve as a promising probe and therapeutic agent for mtRNA, and holds the potential for treating colorectal cancer.

## 2. Results

### 2.1. Synthetic Route of Lead Compound ***QUCO-1***

The synthetic routes of **QUCO-1** are described in Figure 1. Compound **1** was synthesized according to a previous report [28]. A solution of **1** (0.10 g, 0.56 mmol) in acetonitrile (0.16 mL) was treated with CH_3_I (0.1 mL, 1.61 mmol). The mixture was stirred at reflux for 12 h. After cooling, the mixture was filtered and the crude product was thoroughly washed with cold anhydrous ether and dried under a vacuum to afford **2** (0.13 g, yield 73%). A mixture of **2** (0.13 g, 0.41 mmol), 7-*N*,*N*-diethylaminocoumarin-3-aldehyde (0.15 g, 0.62 mmol), and EtOH (3.7 mL) was stirred at reflux for 24 h. After cooling to room temperature, the mixture was filtered and the crude product was thoroughly washed with cold anhydrous ethanal and dried under a vacuum to afford **3** (0.15 g, yield 67%). To a solution of **3** (0.15 g, 0.27 mmol) in acetonitrile (0.75 mL), *N*-methylpiperazine (0.06 mL, 0.54 mmol) and a catalytic amount of K_2_CO_3_ (0.07 g, 0.54 mmol) were added. The reaction mixture was stirred at room temperature for 24 h. After that, the mixture was filtered and the crude product was thoroughly washed with cold anhydrous ethanol and dried under a vacuum to afford a purple solid (**QUCO-1**, 0.11 g, yield 67%).

### 2.2. ***QUCO-1*** as a Fluorescent Probe for mtRNA

Upon screening our probe library, we discovered that **QUCO-1** could image mitochondria (Figure 1 and Appendix A). The probe colocalized well with the mitochondria marker MitoTracker Green in both RKO cancer cells and NCM460 normal cells, with Pearson’s R-values of 0.90 and 0.88, respectively. Given the lipophilic quinolinium cation [29] of **QUCO-1**, it was unsurprising that this molecule was able to accumulate in mitochondria.

**QUCO-1** was derived from RNA fluorescent probes. To verify the ability of **QUCO-1** to interact with RNA in mitochondria, we conducted a fluorescence titration assay. This assay assessed the binding potential between **QUCO-1** and extracted mtRNA and mtDNA. The optical properties of **QUCO-1** are summarized in Appendix A. As depicted in Figure 2, the fluorescence of **QUCO-1** strongly increased in the presence of mtRNA extracted from mitochondria, whereas a modest change was observed with mtDNA. The binding affinity was also calculated, and **QUCO-1** demonstrated a good binding affinity with mtRNA. These findings suggest that **QUCO-1** has the potential to interact with mtRNA in live cells and emit fluorescence.

### 2.3. ***QUCO-1*** Inhibits Cancer Cell Proliferation by Causing Mitochondrial Dysfunction and OXPHOS Inhibition

The presence of **QUCO-1** may alter the function of mtRNA, leading to changes in the overall function of the mitochondria. These changes in mitochondrial function can have a significant impact on cell activity. To evaluate the effect of **QUCO-1** on cancer and normal cells, we selected two cell lines: RKO, a colorectal cancer cell line; and NCM460, a normal human colon mucosal epithelial cell line. As depicted in Figure 3, **QUCO-1** displayed cytotoxicity against RKO, with an IC_50_ value of 1.3 μM. In contrast, **QUCO-1** had a weaker effect on NCM460, with an IC_50_ value 5 times weaker. These results suggest that **QUCO-1** has a selective inhibitory effect on the viability of cancer cells. Interestingly, although **QUCO-1** accumulated in mitochondria in both normal and tumor cells, the compound only exerted a relatively strong toxicity on normal cells. This suggests that there may be differences in the mtRNA status and the way the mitochondria function is regulated between normal and tumor cells.

With the discovery that **QUCO-1** could bind to mtRNA, we sought to evaluate its impact on mitochondrial functions in RKO cells as a potential anticancer agent. We measured two key indicators of the mitochondrial function: the mitochondrial transmembrane potential (MMP), and the reactive oxygen species (ROS) level (Figure 4A,B). MMP is crucial for maintaining the proper function of the mitochondrial respiratory chain whilst an increase in ROS production is a hallmark of mitochondrial dysfunction. After 24 h of treatment, the fluorescence levels of the MMP dyes significantly decreased in the presence of **QUCO-1**, indicating a substantial loss of depolarization in the mitochondrial transmembrane. Meanwhile, the ROS level increased in the presence of **QUCO-1**. These changes in both MMP and ROS levels suggest that **QUCO-1** leads to severe mitochondrial damage.

Given the impact that **QUCO-1** had on mitochondrial dysfunction in cancer cells, we wanted to explore whether it also affected the mitochondrial respiration function in the OXPHOS system. An Agilent Seahorse XF Cell Mito Stress Test Kit was used to measure the oxygen consumption rate (OCR) and extracellular acidification rate (ECAR) of live RKO cells treated with **QUCO-1** (Figure 4C and Appendix A). After 24 h of incubation with **QUCO-1**, the OCR of the RKO cells decreased in a concentration-dependent manner for basal respiration, proton leak, maximal respiration, spare respiration capacity, and ATP-linked respiration. Nonmitochondrial oxygen consumption was not affected. Additionally, we observed particularly severe damage to basal and ATP-linked respiration. In contrast, the ECAR and all glycolysis indicators were not significantly affected (Appendix A). These results, consistent with the trends observed in the MMP and ROS levels, confirmed that **QUCO-1** could cause severe mitochondrial respiration dysfunction. Overall, **QUCO-1** could induce mitochondrial dysfunction, possibly related to OXPHOS, and inhibit cancer cell proliferation.

### 2.4. ***QUCO-1*** Induces OXPHOS Dysfunction and Apoptosis

The mtRNA probe **QUCO-1** could bind mtRNA, induce mitochondrial dysfunction, and cause OXPHOS inhibition. To verify the mechanism, we then determined the changes in OXPHOS proteins by Western blots in the presence of this probe. As shown in Figure 5, the OXPHOS protein levels significantly changed with **QUCO-1** treatment; we noticed that complex II decreased at 0.25 μM. However, complex II increased at 0.5 μM, possibly by unexpected feedback. Overall, it was reasonable that **QUCO-1** could inhibit the expression of mitochondrial respiratory chain complexes and regulate OXPHOS by targeting mtRNA.

Dysfunction of mitochondria usually induces apoptosis [30,31]. Thus, we then determined the hallmark of apoptosis, including BAX/BCL2 and CL-PARP/PARP, by Western blotting. As shown in Figure 6, the level of BAX and BCL2 decreased, and the level of cleavage PARP increased. The phenomenon indicated that **QUCO-1** could induce apoptosis. In conclusion, **QUCO-1** could suppress OXPHOS and severe mitochondrial respiration induced a deficient energy supplement, eventually causing apoptosis.

### 2.5. ***QUCO-1*** Arrests the Cell Cycle and Inhibits Colony Formation

Inspired by the activity of **QUCO-1**, we then evaluated its impact on the cell cycle distribution and colony formation because OXPHOS could arrest the cell cycle and inhibit cell proliferation. After 48 h of being treated with **QUCO-1**, the cells in the G2/M ratio dependently increased with an increasing concentration, which showed that the cells were finally arrested in the G2/M phase (Figure 7A). Moreover, a colony formation assay demonstrated that **QUCO-1** could significantly inhibit RKO colony formation at low concentrations (Figure 7B). In general, QUCO-1 effectively blocked the cell cycle and inhibited the proliferation of tumor cells, both of which indicate that this compound has a good anticancer activity.

## 3. Discussion

mtRNA is a vital component of mitochondria, but few small molecules targeting mtRNA have been reported for its study. By screening our RNA fluorescent probe library, we discovered a fluorescent quinolinium derivative, **QUCO-1**, which targeted mtRNA. This compound could visualize mitochondria by lighting up with mtRNA, making it a promising mtRNA probe. **QUCO-1** also inhibited cancer cell proliferation, suggesting a new strategy to modulate mitochondrial function. Interestingly, although **QUCO-1** could accumulate in mitochondria in both normal and tumor cells, the compound only exerted a relatively strong toxicity on normal cells. This suggests that there may be differences in the mtRNA status and the way the mitochondria function is regulated between normal and tumor cells.

A further evaluation showed that **QUCO-1** could induce severe mitochondrial dysfunction and interfere with OXPHOS whilst not affecting glycolysis. The compound suppressed the expression of mitochondrial respiratory chain complexes by targeting mtRNA, suggesting a relationship between mtRNA and OXPHOS. Although we found that **QUCO-1** could target mtRNA and suppress the expression of mitochondrial respiratory complexes, it is still unclear how **QUCO-1** affected mtRNA and caused changes in the protein expression. We desire to obtain more small molecules with a better specificity to mtRNA, and a structure–activity relationship based on further derivatization is needed in the future.

In this study, we discovered a novel lead mtRNA targeting agent, **QUCO-1**, which could act as a monitor and modulator of mtRNA. The anticancer activity of **QUCO-1** offers a promising strategy for treating colorectal cancer by modulating mtRNA and suppressing the OXPHOS function. The discovery of **QUCO-1** is a good start to a novel anticancer strategy targeting mtRNA.

## 4. Materials and Methods

### 4.1. Chemistry

All analytical grade chemicals purchased were utilized without further purification. ^1^H and ^13^C NMR spectra were recorded using tetramethylsilane (TMS) as the internal standard in DMSO-*d*_6_, Methanol-*d*_4_, or Chloroform-*d* with a Bruker Avance III spectrometer at 400 MHz or 500 MHz. High-resolution mass spectra (HRMS) were recorded on a Shimadzu LCMS-IT-TOF. All synthesized compounds were purified by using flash column chromatography with silica gel (200–300 mesh). The purities of all synthesized compounds were confirmed to be higher than 95% by using an analytical HPLC equipped with a Shimadzu LC-20AB system with an AnalaRic C18 column (4.6 × 250 mm, 5 μm), which was eluted with methanol and water containing 0.1% trifluoroacetic acid at a flow rate of 0.5 mL/min.

*7-Chloro-1,2-dimethyl-1,8-naphthyridin-1-ium iodide* (**2**). ^1^H NMR (400 MHz, DMSO-*d_6_*) *δ* 9.22–9.15 (m, 1H), 8.95 (dd, *J* = 8.5, 3.2 Hz, 1H), 8.26 (dd, *J* = 8.4, 2.9 Hz, 1H), 8.20 (dd, *J* = 8.0, 2.6 Hz, 1H), 4.46 (d, *J* = 2.7 Hz, 3H), 3.11 (d, *J* = 2.7 Hz, 3H). LRMS (ESI): found 193.05.

*(E)-7-Chloro-2-(2-(7-(diethylamino)-2-oxo-2H-chromen-3-yl)vinyl)-1-methyl-1,8-naphthyridin-1-ium iodide* (**3**). ^1^H NMR (400 MHz, DMSO-*d_6_*) *δ* 8.96 (d, *J* = 8.8 Hz, 1H), 8.80 (d, *J* = 8.4 Hz, 1H), 8.65 (d, *J* = 9.0 Hz, 1H), 8.50 (s, 1H), 8.30 (d, *J* = 15.4 Hz, 1H), 8.08 (t, *J* = 11.6 Hz, 2H), 7.62 (d, *J* = 9.0 Hz, 1H), 6.88 (d, *J* = 8.5 Hz, 1H), 6.69 (s, 1H), 4.45 (s, 3H), 3.55 (d, *J* = 6.8 Hz, 4H), 1.18 (t, *J* = 6.8 Hz, 6H). LRMS (ESI): found 420.15.

*(E)-2-(2-(7-(Diethylamino)-2-oxo-2H-chromen-3-yl)vinyl)-1-methyl-7-(4-methylpiperazin-1-yl)-1,8-naphthyridin-1-ium iodide* (**QUCO-1**). Following the mentioned method, the compound **QUCO-1** was obtained as a purple solid (0.15 g, 45%). ^1^H NMR (400 MHz, DMSO-*d*_6_) *δ* 8.58 (d, *J* = 8.5 Hz, 1H), 8.38 (s, 1H), 8.29 (d, *J* = 9.1 Hz, 1H), 8.03 (d, *J* = 8.6 Hz, 1H), 7.97 (s, 1H), 7.84 (d, *J* = 15.7 Hz, 1H), 7.58 (d, *J* = 9.4 Hz, 2H), 6.83 (d, *J* = 9.2 Hz, 1H), 6.65 (s, 1H), 4.30 (s, 3H), 3.93 (s, 4H), 3.59–3.45 (m, 4H), 3.34 (s, 3H), 2.27 (s, 4H), 1.16 (t, *J* = 6.9 Hz, 6H). ^13^C NMR (126 MHz, DMSO-*d_6_*) *δ* 160.21 (s), 159.11 (s), 156.95 (s), 154.78 (s), 152.75 (s), 149.55 (s), 146.66 (s), 141.89 (s), 139.97 (s), 139.31 (s), 131.47 (s), 118.50 (s), 117.47 (s), 115.29 (s), 113.84 (s), 112.76 (s), 110.73 (s), 108.93 (s), 96.73 (s), 54.73 (s), 45.90 (s), 44.92 (s), 35.68 (s), 12.90 (s). Purity: 95.673% by HPLC. HRMS (ESI): calcd for (M − I)^+^ (C_29_H_33_N_5_O_2_^+^) 484.2707: found 484.2680.

### 4.2. Cell Culture

RKO cells (Procell, CL-0196, Wuhan, China) and NCM460 cells (Procell, CL-0196, Wuhan, China) were grown in a Roswell Park Memorial Institute 1640 medium (RPMI 1640 medium) (Cienry, CR31800, Huzhou, China) complemented with 10% fetal bovine serum (FBS) (ExCell Bio, FSP500, Taicang, China). A 1% penicillin–streptomycin solution (P/S) (Procell, PB180120, Wuhan, China) was added. All cells were cultured at 37 °C in a constant temperature incubator with 5% CO_2_.

### 4.3. Cytotoxicity Assay

The cytotoxicity was measured using the Cellomics ArrayScan Vti high-content imaging platform (Thermo Fisher Scientific, Waltham, MA, USA). RKO and NCM460 cells were seeded in a 96-well plate (5000 cells/well). After adhering overnight, they were then treated with various concentrations of **QUCO-1** in a normal RPMI 1640 medium for 48 h. After treatment, the cells were fixed with 70% ethanol overnight, then stained with propidium iodide (PI) (BioFroxx, PI-1246MG010, Einhausen, Germany) at room temperature for over 2 h. A high-content imaging platform was set to automatically focus on the PI fluorescence channel, and determined the cell density by counting the nucleus number in each well. IC_50_ values were calculated from the curves of the cell density values of the triplicate tests plotted against the compound concentration.

### 4.4. Cell Cycle Analysis

For the cell cycle analysis, RKO cells were seeded in a 24-well plate (20,000 cells/well). After adhering overnight, they were then treated with different concentrations of **QUCO-1** for 48 h. After treatment, the cells were fixed with 70% ethanol overnight, then stained with propidium iodide (PI) (BioFroxx, PI-1246MG010, Einhausen, Germany) at room temperature for over 2 h. The PI intensity representing the DNA amounts in the individual cells was automatically quantified, and the cell cycle distribution was analyzed by HCS Studio Cell Analysis software 2.0.

### 4.5. Colocalization

RKO cells were seeded in 96-well glass-bottom plates (cellvis, P96-1-N) (5000 cells/well) and allowed to adhere overnight. **QUCO-1** (final concentration: 1 μM) and MitoTracker Green (E_x_ = 490 nm, E_m_ = 523 nm, Yeasen, 40743ES50) (final concentration: 100 mM) were co-stained for 1 h and then observed using laser scanning confocal microscopy, followed by excitation wavelengths of 561 nm (QUCO-1) and 488 nm (MitoTracker Green) and detection wavelengths of 570–620 nm (QUCO-1) and 500–540 nm (MitoTracker Green). The colocalization analysis was performed using the Coloc 2 plugin of ImageJ version 1.53c.

### 4.6. Colony Formation Assay

RKO cells were seeded in 6-well plates (500 cells/well) and exposed to QUCO-1 with the RPMI 1640 medium at 37 °C in a 5% CO_2_ incubator for 7 days. The cells were fixed with 4% polyoxymethylene in PBS for 15 min and dyed with crystal violet.

### 4.7. OCR and ECAR Determination

RKO cells were seeded in XF 96-well cell culture microplates in quintuplicate at 5000 cells/well in 100 μL of the cell culture medium (RPMI 1640 with 10% FBS) and then incubated at 37 °C in 5% CO_2_ for 24 h. Every group experiment was carried out in 6 replicates. The cell culture medium was changed to a new medium with or without **QUCO-1** for 24 h. First, the growth medium from each well was removed and replaced with 100 μL of a certain assay medium pre-warmed to 37 °C. The cells were incubated at 37 °C immediately for 30 min to reach the temperature and pH equilibrium before the first rate measurement. Before the measurement, the XF-96 Analyzer gently mixed the certain medium in each well for 10 min to reach an equilibrium state of oxygen partial pressure. Following mixing, the OCR and ECAR were simultaneously measured for 5 min to establish a baseline. The certain medium was then gently mixed again for 5 min between each rate measurement to restore the normal oxygen tension and pH equilibrium in the cell microenvironment. After the baseline measurement, 20 μL of a testing agent dissolved in an assay medium was injected into each well to reach the desired final working concentration, followed by mixing for 5 min to expedite the compound exposure to the targeting proteins, after which both OCR and ECAR measurements were then made. Three baseline data and three response data were measured after each compound addition (1 μM Oligomycin, 2 μM FCCP, Rotenone for OCR; 10 mM Glucose, 1 μM Oligomycin, 50 mM 2-DG for ECAR), and the average of all baseline data or test data was used for the data analysis. The data were normalized by the quantitation of the total protein of cells in each well.

### 4.8. Mitochondria Membrane Potential Determination

RKO cells were planted in 6-well plates (200,000 cells/well) for 24 h. The RKO cells were then treated with various concentrations of **QUCO-1** for 24 h and harvested in RPMI 1640 without FBS. Cells treated with **QUCO-1** were detected using Rhodamine 123 (final concentration: 2 μΜ). After incubation for 30 min at 37 °C, the cells were resuspended with RPMI 1640. Flow cytometry was used to analyze two indicators by using a CytoFLEX S flow cytometer (Beckman Coulter, Brea, CA, USA). For each analysis, 2 × 10^4^ events were collected. The cell cycle distribution was analyzed using FlowJo software v8.

### 4.9. Determination of Mitochondria and ROS Determination

RKO cells were planted in 6-well plates (200,000 cells per well) for 24 h. The RKO cells were then treated with various concentrations of **QUCO-1** for 24 h and harvested in RPMI 1640 without FBS. Finally, the mitochondria ROS level was detected by using the guide of a BBoxiProbe^®^ O08 probe.

### 4.10. Determination of Protein Extract and Western Blot Assay

After treatment with **QUCO-1** for 48 h, the RKO cells were washed with cold PBS three times and lysed with a RIPA lysis buffer (50 mM Tris, 150 mM NaCl, 1% TritonX-100, 1% sodium deoxycholate, 0.1% SDS, and 1 mM EDTA; pH 7.4) at 4 °C for 30 min. The lysis of cells was centrifuged at 12,000 rpm at 4 °C for 15 min. The quantitation of the total protein of the supernatant was diluted for the BCA determination and denatured at 85 °C for 10 min with the addition of a loading buffer (50 mM Tris-HCl, 6 M urea, 6% 2-mercaptoethanol, 3% SDS, and 0.003% bromophenol blue; pH 6.8). A total of 20 μg of protein was loaded onto each lane, run on the 5%/12% SDS-PAGE, transferred to microporous polyvinylidene difluoride membranes, and then detected by Western blots. The primary antibody used in this experiment was Total OXPHOS Human WB Antibody Cocktail (ab110411, abcam). The secondary antibodies in this assay were horseradish peroxidase-conjugated anti-mouse (No. 7076S, Cell Signaling Technology) and anti-rabbit (No. 7074S, Cell Signaling Technology). The visualization of the protein bands replied on the chemiluminescence substrate.

### 4.11. Mitochondrial RNA Assay and DNA Extraction

After treatment with **QUCO-1** for 24 h, the RKO cells were washed three times with cold PBS and the mitochondria were extracted using a cell mitochondrial isolation kit (beyotime, C3601). RNA and DNA extractions were performed according to the instructions of the RNAeasy™ animal RNA extraction kit (beyotime, R0027) and the genomic DNA small extraction kit (beyotime, D0063). Finally, their concentrations were determined using trace ultraviolet light after extraction.

### 4.12. Fluorescence Spectrometric Titrations

The fluorescence titration was performed using a fluorophotometer. The mtRNA and mtDNA from the extraction were adjusted to 200 ng/μL using enzyme-free ddH_2_O. The compound **QUCO-1** (excitation wavelength of 540 nm) solution was configured using enzyme-free ddH_2_O with a concentration of 5 μM. mtRNA/mtDNA was successively added (50 ng each) into the compound solution and mixed before detection using fluorescence spectrometry. After using originlab for the baseline processing, the regression processing of the peak data and the K_d_ calculation were performed using GraphPad 7.0.

## Data Availability

Data are contained within the article or Appendix A.

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
