# Peer review of "Fluorescent Quinolinium Derivative as Novel Mitochondria Probe and Function Modulator by Targeting Mitochondrial RNA"

_molecules, 2023, doi:10.3390/molecules28062690_

Round 1
Reviewer 1 Report
In this manuscript, Wang et al present a mitochondria-targetable quinolinium derivative enabling to light up mitochondria and induce severe mitochondrial dysfunction by binding with MtRNA. This work is meaningful and well-organized. I recommend it can be considered for publication after address the following issues:
1. The representation of the probe's linear relationship in Figure 2A is inadequate and the illustration appears to be crudely drawn.
2. Co-localization experiments and fluorescence titration assays had been carried out to verify the affinity and specificity targeting of QUCO-1 for mtRNA extracted from mitochondria in cells. However, it is unclear whether the probe would bind to mtRNA in both normal and RKO cells, and whether this would cause issues with probe dispersion in practical applications. The related data is lacking.
3. The validation experiments conducted on mtRNA and mtDNA were solely based on in vitro spectroscopy, and additional selective experiments are necessary to provide further evidence for the probes' efficient mitochondrial targeting.
4. Data of binding constants should be provide during fluorescence titration assays on mtRNA and mtDNA extracted from mitochondria in cells to verify QUCO-1's affinity for mtRNA.
5. In MMP and ROS experiments, the author could include an additional set of data on the effects of QUCO-1 on normal cells to increase the overall rigor of the data.
6. The probe exhibits significant cytotoxicity, and in Figures 8A and B, the experiments were solely conducted on RKO cells without including control groups of normal cells.
Author Response
We are so grateful for your positive review opinion and constructive suggestions. We deeply value the feedback and have taken great care to address all concerns raised. A point-by-point response to the comments is listed below in the file.

Reviewer 2 Report
In this manuscript, the authors design and synthesis of a fluorescent quinolinium derivative, QUCO-1, that has been shown to light up with mtRNA in vitro and live cells. The fluorescent probe QUCO-1 could colocalize well with the mitochondria marker Mito-Tracker Green, with a Pearson's R-value of 0.90 and inhibit cancer cell proliferation by causing mitochondrial dysfunction and OXPHOS inhibition. The experimental procedures and the results of fluorophore seem technically sound, but seems containing some ambiguous or unclear interpretation of the experimental results both in fluorophore design and biological assessment. The detailed comments are listed below.
1. In Figure 5, the fit line of QUCO-1 (treated for 24 h) on RKO cells in galactose-medium have some question. If the author consider that RKO cells showed significantly higher sensitivity in galactose-medium, the fit line of galactose-medium should be steeper than fit line of glucose-medium rather than gentle. Please recheck the data and fit.
2. In Figure 5, the content of complex â…¡-SHDB decreased significantly at 0.25μM and then increased on 0.5μM QUCO-1 treated. Please give a reasonable explanation or conduct the experiment again.
3. In Figure 5, please supply western blotting analysis of apoptosis-related protein treatment with the concentration of QUCO-1 at 1 μM and 2μM.
4. In Figure 2 B and D, the unit of abscissa mtRNA may be wrong. And In page 2, line 61, there miss a full point. Please recheck the sentences carefully.
Author Response
We would like to thank the reviewer for the comments and suggestions. We appreciate you had a high opinion of our novelty of findings. We deeply value the feedback and have taken great care to address all concerns raised. A point-by-point response to the comments is listed below in the file.

Reviewer 3 Report
In this submitted manuscript, Chen and co-workers are reporting an interesting Quinolinium-based hemicyanine dye for sensing mitochondria RNA and thereby for numerous biological sensing applications via live cell fluorescence microscopy. Probe also exhibited distinguishable differences in its activity in cancer vs healthy cells. This probe therefore can be highly useful for developing interesting fluorescence probe platforms for bioimaging applications. However, there are several improvements must be performed for this submitted version to improve the scientific soundness of this work. Specifically following suggestions should be performed prior to the acceptance of this work.
(1). The abstract of the manuscript must be improved while paying close attention to the textual and grammatical mistakes. Abstract should clearly highlight the significance of the current findings and must remove general statements.
(2). In the introduction 2nd paragraph (line 33-40) authors discuss about recent development of small molecule probes towards sensing ETC. In this paragraph authors should also discuss briefly about the current history of developing fluorescent probes towards detecting mitochondrial membrane potential dysfunction. Some of the most recently developed probes should be references. For example : Coordination Chemistry Reviews, 420, 213419; ChemBioChem 23, no. 2 (2022): e202100516; Sensors and Actuators B: Chemical 327 (2021): 128929; Bioorganic Chemistry 99 (2020): 103848; Sensors and Actuators B: Chemical 292 (2019): 16-23.
(3). Authors should summarize optical properties of the fluorescent probe in different solvent systems and must be reported in the manuscript as a table format. This should include, absorbance, emission, Stokes’ shift, molar absorptivity and the fluorescence quantum yield.
(4). Authors should provide 13-C NMR data for compounds # 2-3.
(5). Under the fluorescence microscopy images (figure 1) please indicate what is the staining concentration clearly.
(6). According to the provided data, the IC50 value in RKO cells has a value of 1.3 µM. However, the staining concentration for the live cells is 1 µM according to the fluorescence microscopy protocol. This staining concentration can significantly affect cellular activity during the imaging. What is authors opinion on this?
(7). Please calculated the LOD value in figure 2B.
(8). Section 2.5 which explains the impact towards cell cycle and colony formation must be improved to elaborate the experimental protocol and the findings.
(9). The title of the section 4.13 must be corrected as “Fluorescence spectrometric titrations”.
(10). Authors must also include the bright-field view in the fluorescence microscopy images if possible.
Author Response
Thank you for your time involved in reviewing the manuscript. It is an honor for us to get so kindly review opinions for you. We deeply value the feedback and have taken great care to address all concerns raised. A point-by-point response to the comments is listed below in the file.

Round 2
Reviewer 2 Report
The authors have solved my questions.